# The Incidence of Sport-Related Anterior Cruciate Ligament Injuries: An Overview of Systematic Reviews Including 51 Meta-Analyses

**DOI:** 10.3390/jfmk10020174

**Published:** 2025-05-14

**Authors:** Javier Martinez-Calderon, Marta Infante-Cano, Javier Matias-Soto, Veronica Perez-Cabezas, Alejandro Galan-Mercant, Cristina Garcia-Muñoz

**Affiliations:** 1Departamento de Fisioterapia, Universidad de Sevilla, 41009 Sevilla, Spain; jmcalderon@us.es; 2CTS 1110: Uncertainty, Mindfulness, Self, and Spirituality (UMSS) Research Group, 41009 Andalusia, Spain; martainfante0@gmail.com (M.I.-C.); msjavi93@gmail.com (J.M.-S.); ccriss.g@gmail.com (C.G.-M.); 3Cochrane Rehabilitation, Functioning and Disability, London W1G 0AN, UK; 4Instituto de Biomedicina de Sevilla-IBiS (Hospitales Universitarios Virgen del Rocío y Macarena, CSIC, Universidad de Sevilla), 41013 Sevilla, Spain; 5Departamento Ciencias de la Salud y Biomédicas, Universidad Loyola Andalucía, 41074 Sevilla, Spain; 6Department of Nursing and Physiotherapy, University of Cadiz, 11009 Cadiz, Spain; alejandro.galan@uca.es; 7Biomedical Research and Innovation Institute of Cadiz (INIBICA), 11009 Cadiz, Spain; 8MOVE-IT Research Group, Department of Physical Education, Faculty of Education Sciences, University of Cadiz, 11009 Cadiz, Spain

**Keywords:** ACL, incidence, injury, overview, review, sports

## Abstract

**Background/Objectives:** The number of systematic reviews evaluating the incidence of anterior cruciate ligament (ACL) injuries in sports is increasing. To synthesize pooled incidence and prevalence rates of sport-related ACL injuries based on published systematic reviews with meta-analyses. **Methods:** An overview of systematic reviews with meta-analysis was conducted. The CINAHL, Embase, PubMed, and SPORTDiscus databases were searched from inception to 17 October 2023. AMSTAR 2 was used to assess the methodological quality of reviews. The degree of overlap between reviews was calculated when possible. **Results:** Seven systematic reviews including 51 meta-analyses of interest were included. The prevalence of ACL injuries was not meta-analyzed. Meta-analyses mainly showed that ACL injuries may have a high incidence in American football, basketball, European football/soccer, and volleyball, among other sports. In addition, ACL injuries may have a higher incidence in females than males in some sports. For example, the pooled incidence rates of ACL injuries in basketball ranged from 0.091 (95%CI, 0.074–0.111) to 0.110 (95%CI, 0.094–0.128) among female athletes, whereas this incidence ranged from 0.024 (95%CI, 0.016–0.034) to 0.027 (95%CI, 0.019–0.035) among male athletes. **Conclusions:** Sport-related ACL injuries may have a high incidence in sports such as American football, basketball, European football/soccer, or volleyball and show differences between sexes. Therefore, a sex-specific prevention of these injuries may be needed.

## 1. Introduction

The economic and societal impact of sport-related ACL injuries is substantial at all levels of competition. Return to sport could be delayed after ACL injuries such as ACL tears, which may cause important economic damage for athletes and their teams [1]. The cumulative economic loss from ACL injuries in the National Basketball Association (NBA) was $99 million from 2000 to 2015 [2]. In addition, players from the National Football League (NFL) with ACL injuries have reported losing more than $2 million in comparison to NFL players without these injuries [3]. ACL injuries are also the second musculoskeletal injury with a major economic burden after hamstring injuries in professional European footballers [1], who have reported economic losses estimated at €84.499 [1].

Athletes with ACL injuries aim to return to sport as soon as possible [4]. However, some challenges can appear during their recovery. For example, the expectations of athletes to return to sport, the confidence of their coaches, and the stress associated with the compliance of contract obligations and sponsorships may be evident [4,5]. Knee osteoarthritis is common after ACL injuries and a third of individuals who undergo an ACL reconstruction may develop it [6]. Furthermore, depression, kinesiophobia, and low levels of self-efficacy are common before ACL surgeries [7,8].

Currently, many systematic reviews have summarized the epidemiology of sport-related ACL injuries around the world [1,9,10,11,12,13,14]. Therefore, we argue that an overview of systematic reviews is timely and adequate to show the status of the meta-analyses in this field [15]. We aimed to conduct an overview of systematic reviews with meta-analysis to summarize the pooled incidence and prevalence rates of sport-related ACL injuries.

## 2. Materials and Methods

We followed the Preferred Reporting Items for Overviews of Reviews (PRIOR) statement [16] and the Preferred Reporting Items for Systematic Reviews and Meta-Analyses (PRISMA) statement for abstracts [17]. We screened PROSPERO, Open Science Framework (OSF), and the International Platform of Registered Systematic Review and Meta-analysis Protocols (INPLASY) databases to ensure that similar overviews were not ongoing or published. Then, we prospectively registered the review protocol in OSF: blinded (Appendix A) https://doi.org/10.17605/OSF.IO/FJN4T.

### 2.1. Deviations from the Review Protocol

Some deviations were conducted to improve the quality of this overview and reach more direct conclusions. We focused this study on athlete populations. ACL injuries are an area of enormous interest in musculoskeletal rehabilitation. Therefore, given the significant number of studies available in this field, and because athletes represent a population with different characteristics than the general population (for example, due to the pressure to return to sports), we decided to focus the study exclusively on athletes to reach more consistent and direct conclusions. Finally, maps of prevalence were not conducted since original articles may have been missed in the included reviews. This was not a protocol deviation. This was a scope adjustment because we did not find meta-analyses of prevalence studies considering ACL injuries in sports.

### 2.2. Data Sources and Search Strategy

One co-author (JMC) searched for the CINAHL (via EBSCOhost), Embase (via Elsevier), PubMed, and SPORTDiscus (via EBSCOhost) databases from inception to 17 October 2023. The following search filters were used when possible: the type of document (e.g., reviews) and the language of publication (English or Spanish). In addition, this co-author aimed to manually search systematic reviews that check our inclusion criteria in overviews of reviews or review protocols related to our scope. Appendix A shows all search strategies.

### 2.3. Eligibility Criteria

One co-author (CGM) used the Patient/Population, Exposure, Comparison, Outcome, Study design (PECOS) framework [18] to develop the eligibility criteria.

Inclusion criteria:

P: Athletes with ACL injuries without restrictions in terms of gender/sex, age, type of sport, or level of competition (amateur or professional).

E: ACL injuries.

C: Not applicable.

O: The pooled incidence (risk over time or exposure) and/or prevalence (existing cases at a point or period) rates of sport-related ACL injuries.

S: Systematic reviews with meta-analysis published in peer-reviewed journals and written in the English or Spanish language. Systematic reviews were included if reviews used systematic and reproducible methods to collect data on primary studies, critically analyze the evidence, and summarize the results quantitatively [16]. Systematic reviews should report the sources where the search strategies were conducted, eligibility criteria, study selection processes and data extraction, and methods to evaluate and synthesize the results. We also considered those systematic reviews that did not assess the methodological quality or the risk of bias in primary studies to analyze all systematic reviews with meta-analysis on our topic.

Exclusion criteria:

[I] Meta-analysis did not evaluate the incidence or prevalence of ACL injuries by specific sports.

[II] Meta-analysis combined ACL injuries with other knee pathologies.

[III] Impossibility of accessing full text. We contacted the corresponding authors via email, but no responses were received.

[IV] Conference abstracts and proceedings.

### 2.4. Study Selection

One co-author (MIC) used Zotero 6.0.9 Citation Management Software to conduct the study selection. This co-author manually screened all references and removed duplicates [19]. Afterward, titles and abstracts were read, and full texts were analyzed when abstracts seemed eligible or when abstracts were unavailable. No consensus was needed.

### 2.5. Methodological Quality Assessment

Two independent co-authors (VPC and AGM) used AMSTAR 2 to assess the methodological quality of reviews [20]. AMSTAR 2 has 16 items that can be rated as yes, partially yes, or no. The overall score is not recommended [20], but seven items are considered critical: items: 2, 4, 7, 9, 11, 13, 15 [20]. Disagreements between these co-authors were solved by consensus. The percentage of agreement between co-authors was calculated using the number of items rated with the same score before pooling the results of their independent assessments. The items associated with interventions and/or comparator groups were not applied or were adjusted to observational studies (e.g., items 1, 3, 8, 9, or 11).

### 2.6. Data Extraction

One co-author (JMS) extracted from the included reviews if the authors conducted the following analyses: meta-regressions, sensitivity analyses, subgroup meta-analyses, the methodological quality or the risk of bias assessment, and the use of the Grading of Recommendations Assessment, Development, and Evaluation (GRADE) system to rate the certainty of evidence. The following information was also extracted: first author and year of publication, outcomes, type of sport, the number of studies meta-analyzed, gender, ACL injury cases over the number of athlete exposures or hours of exposure, and main findings. Corresponding authors were contacted via email to request or clarify some information if needed.

### 2.7. Data Synthesis

One co-author (JMS) reported the results in the main text by gender, type of sport, and the way incidence or prevalence was shown (1000 athlete-exposures, 10,000 athlete-exposure, or hours of exposure). One co-author (MIC) developed matrices of evidence to calculate the corrected covered area (CCA). The CCA was calculated to assess the degree of overlap between reviews [21]. The CCA refers to the area that is covered after removing primary studies the first time they are counted. The CCA is calculated based on three aspects: N is the total number of original studies (including duplicates) in the meta-analyses of interest (the sum of all checked boxes in the citation matrix). Furthermore, r is the number of original studies without accounting for duplicates. Finally, c is the number of systematic reviews included in the evidence matrix. The overlap can be classified as slight (CCA 0–5%), moderate (CCA 6–10%), high (CCA 11–15%), or very high (CCA > 15%) [21]. The degree of overlap was only calculated if at least two reviews meta-analyzed the same sport and the same epidemiological parameter (incidence or prevalence). American football and rugby show many similarities. Therefore, we included in the same matrix of evidence both sports. The situation also occurred with European football and soccer. One co-author (JMC) developed upset plots to depict the CCA values.

## 3. Results

A total of 637 references were retrieved from the electronic databases. No extra studies were retrieved by other methods (e.g., manual searches). After removing duplicates, 288 references were read by title and abstract, and 51 full texts were analyzed. Seven systematic reviews with meta-analysis were [1,9,10,11,12,13,14]. Two corresponding authors were contacted via email to request the full text of their reviews, but no responses were received. Therefore, these reviews were excluded for the reason “impossibility of accessing full text”. Figure 1 shows the PRISMA 2020 flow diagram.

### 3.1. Main Characteristics of the Included Reviews

No reviews meta-analyzed the prevalence of sport-related ACL injuries. Incidence rates were reported per 1000 athlete-exposures [9,10,11,13,14], 10,000 athlete-exposures [12], and 1000 h of exposure [1,10]. One review did not specify the number of hours of exposure [9]. The sports explored were American football [9,10,11,13], baseball/softball [9], basketball [9,10,11,13,14], European football/soccer [1,9,10,11,12,13], field hockey [9], floorball [10], lacrosse [9], rugby [9,10,13], volleyball [9,11,13], and wrestling [9]. Five out of seven reviews evaluated elite/professional athletes [1,10,12,13,14]. No reviews used GRADE to rate the certainty of evidence. One review conducted meta-regressions [10]. Two reviews carried out sensitivity [12,14] and all reviews conducted subgroup meta-analyses.

### 3.2. The Degree of Overlap Between Reviews

The overlap between reviews meta-analyzing American football or rugby was classified as moderate (k = 5, CCA = 9%). The overlap was classified as high among reviews meta-analyzing European football or soccer (k = 6, CCA = 12%), and very high between reviews meta-analyzing basketball (k = 5, CCA = 24%) and volleyball (k = 3, CCA = 83%) (Figure 2, Figure 3, Figure 4 and Figure 5). Specifically, it is important to underline very high overlap between systematic reviews meta-analyzing incidence studies in basketball and volleyball. This implies that some original research may be present in meta-analyses from different systematic reviews and, thus, the conclusions of these reviews may exist an overlap in the conclusions of these reviews.

### 3.3. Methodological Quality Assessment

Overall, systematic reviews did not check four critical items of AMSTAR 2. Most of the reviews did not prospectively register their protocol or did not meet the minimum criteria proposed by AMSTAR 2 (item 2). Search strategies were not sufficiently detailed (item 4). A list of excluded studies with the reasons for their exclusion was not provided (item 7). No interpretation or discussion regarding how the quality of primary studies could impact the meta-analyzed results (item 13). In addition, all reviews did not report on the funding sources of primary studies (item 10). The inter-rater agreement was 89%.

#### 3.3.1. Incidence of ACL Injuries in American Football

The pooled incidence rates among American football male players per 1000 athlete-exposures ranged from 0.06 (95%CI, 0.05–0.08) to 0.101 (95%CI, 0.092–0.111) [9,10,11,13], and was 0.055 (95%CI, 0.039–0.074) per hour of exposure [9].

#### 3.3.2. Incidence of ACL Injuries Among Baseball/Softball Athletes

The pooled incidence rate per 1000 athlete exposures was 0.028 (95%CI, 0.017–0.042) among female softball athletes and 0.010 (95%CI, 0.005–0.018) among male baseball athletes [9].

#### 3.3.3. Incidence of ACL Injuries in Basketball

The pooled incidence rates per 1000 athlete-exposures ranged from 0.091 (95%CI, 0.074–0.111) to 0.110 (95%CI, 0.094–0.128) among female athletes and ranged from 0.024 (95%CI, 0.016–0.034) to 0.027 (95%CI, 0.019–0.035) among male athletes [9,11]. On the other hand, two reviews did not report a global pooled incidence rate per 1000 athlete exposures and showed the pooled incidence rates for levels of competition [13,14]. The pooled incidence rates ranged from 0.09 (95%CI not reported) to 0.102 (95%CI, 0.080–0.125) among female high-school athletes and it was around 0.02 among male high-school athletes. The pooled incidence rates ranged from 0.246 (95%CI, 0.219–0.272) to 0.29 (95%CI not reported) among female college athletes and ranged from 0.077 (95%CI, 0.071–0.084) to 0.08 (95%CI not reported) among male college athletes. The pooled incidence rates were 0.20 (95%CI not reported) among female professional athletes and 0.21 (95%CI not reported) among male professional athletes. Finally, one review combined female and male athletes in the same meta-analysis, and no independent results by gender were reported [10]. Finally, the pooled incidence rates per hour of exposure were 0.067 (95%CI, 0.048–0.092) among female athletes and 0.024 (95%CI, 0.006–0.060) among male athletes [9].

#### 3.3.4. Incidence of ACL Injuries in European Football/Soccer

The pooled incidence rates per 1000 athlete-exposures ranged from 0.116 (95%CI, 0.146–0.189) to 0.32 (95%CI not reported) among female athletes and ranged from 0.040 (95%CI 0.029–0.055) to 0.12 (95%CI not reported) among male athletes [1,9,11,13]. One review combined female and male athletes in the same meta-analysis and no independent results by gender were reported [10]. In addition, one review reported the pooled incidence rates of indoor soccer and found a pooled incidence of 5.21 (95%CI not reported) among female athletes and 1.88 (95%CI not reported) among male athletes [13]. Finally, the pooled incidence rates per 10,000 athlete-exposures was 1.7 (95%CI, 1.4 to 2.1) among female athletes and 0.9 (95%CI, 0.7 to 1.1) among male athletes [12], and the pooled incidence rates per hour of exposure was 0.090 (95%CI, 0.059–0.131) among female athletes and 0.006 (95%CI, 0.000–0.033) among male athletes [9].

#### 3.3.5. Incidence of ACL Injuries in Field Hockey

The pooled incidence rate per 1000 athlete exposures was 0.035 (95%CI, 0.019–0.058) among female athletes [9].

#### 3.3.6. Incidence of ACL Injuries in Floorball

The pooled incidence rates per 1000 athlete-exposures were combined for female and male athletes and no results for genders were reported [10].

#### 3.3.7. Incidence of ACL Injuries in Lacrosse

The pooled incidence rates per 1000 athlete exposures were 0.112 (95%CI, 0.092–0.135) among female athletes and 0.088 (95%CI, 0.066–0.114) among male athletes [9].

#### 3.3.8. Incidence of ACL Injuries in Rugby

The pooled incidence rate per 1000 athlete-exposures was 0.36 (95%CI not reported) among female athletes and ranged from 0.06 (95%CI, 0.02–0.18) to 0.18 (95%CI not reported) among male athletes [10,13] In addition, the pooled incidence rate per hour of exposure was 0.049 (95%CI, 0.030–0.075) among male athletes [9].

#### 3.3.9. Incidence of ACL Injuries in Volleyball

The pooled incidence rates per 1000 athlete-exposures ranged from 0.018 (95%CI 0.010–0.029) to 0.023 (95%CI, 0.014–0.036) among female athletes [9,11]. One review found no ACL injury cases [13].

#### 3.3.10. Incidence of ACL Injuries in Wrestling

The pooled incidence rate per 1000 athlete exposures was 0.034 (95%CI, 0.023–0.049) among male athletes [9].

## 4. Discussion

This overview of systematic reviews with meta-analysis aimed to summarize the pooled incidence and prevalence rates of sport-related ACL injuries by type of sport. Overall, the included meta-analyses mainly found ACL injuries may have a high incidence in American football, basketball, European football/soccer, and volleyball. On the other hand, we did not find reviews meta-analyzing the prevalence of sport-related ACL injuries.

The prevalence of sport-related ACL injuries has been explored in some primary studies. For example, the prevalence of these injuries was 14.7% among Saudi Arabian football athletes (N = 320) [22]. In addition, this prevalence was higher among wakeboarding athletes (N = 123) in the USA, reaching 42.3% [23]. In Turkey, around 15% (N = 31) of individuals who practiced recreational skiing activities had ACL injuries [24]. These studies show the prevalence of sport-related ACL injuries is a topic of interest and needs to be investigated in more countries and sports. We encourage researchers in this field to increase the body of knowledge on this topic to reach more direct conclusions.

Gender/sex-based disparities were observed across the meta-analyses. ACL injuries may have a higher incidence among females than males in sports such as basketball or European football/soccer. Previous studies support our findings. These studies showed female athletes experienced a two- to six-fold higher risk of ACL injury than males [25,26,27]. The interaction of multiple factors may have to explain these results such as the anatomy, biomechanics, or environment surrounding the individual [26,27]. In this context, a new gendered environmental approach has highlighted the role of environmental and contextual factors in the development and progression of ACL injuries [28]. We encourage clinicians and researchers in this field to consider aspects such as the expectations of females and males regarding sports, gender-based risk-taking beliefs, or gender-based inequities in life outside sport [29]. These factors may help explain possible differences in the epidemiology of sport-related ACL injuries.

On the other hand, the stratification of sport-related injuries may be complicated in some countries. In the USA, multiple organizations (different insurance companies, National Collegiate Athletic Association (NCAA), and professional sporting organizations) may provide epidemiology data in sports, which may affect the way the epidemiology of sport-related injuries is studied. However, Scandinavian countries have a nationalized health registry that leads to easy comparisons. Readers should be aware of this situation when interpreting our results.

Furthermore, some methodological aspects need to be discussed. We excluded some reviews because their meta-analyses jointly analyzed primary studies evaluating different types of sports and subgroups were not conducted. For example, Montalvo et al. evaluated 58 studies, but their meta-analyses were not stratified by type of sport [12]. Montalvo et al. also developed another review that analyzed 36 studies [29]. They included subgroup meta-analyses by collision, limited contact, and non-contact sports, but, meta-analyses were not stratified by specific sports [29]. In addition, the degree of overlap between meta-analyses focused on basketball or volleyball was calculated as very high. This is problematic because multiple meta-analyses could evaluate similar original research and sports clinicians and researchers should be aware about this issue when comparing the results of different systematic reviews with meta-analysis. Finally, no reviews used the GRADE system to rate the certainty of evidence [30]. The GRADE system is composed of several domains that are key to understanding the certainty and strength of the meta-analyzed findings. Therefore, sports clinicians and researchers need to know that despite the results provided by meta-analyses, we do not know how the risk of bias, inconsistency, indirectness, imprecision, and publication bias could impact the certainty of the meta-analyzed findings. In this context, we strongly encourage the authors of systematic reviews in this field to use the GRADE approach when developing their reviews. Although there is not a specific GRADE guideline for rating systematic reviews of epidemiological studies, it is possible to use GRADE in this type of reviews since all the GRADE domains (risk of bias, inconsistency, indirectness, imprecision, and publication bias) are covered by epidemiological systematic reviews. We encourage readers to consult a recent editorial that has underlined the importance of using GRADE in epidemiological reviews of sport-related injuries [31].

### 4.1. Future Research

As previously discussed, the number of primary studies evaluating the prevalence of ACL injuries in different types of sports continues to grow. Therefore, future research should aim to synthesize and critically evaluate these primary studies, if possible, through a systematic review with or without meta-analysis, or at least through a scoping review. This information is vitally important for identifying potential gaps in the state of knowledge on this topic, which can guide not only future research but also new health policies in this field.

### 4.2. Limitations

We acknowledged that some systematic reviews may have been missed since only studies published in peer-reviewed journals were considered. In addition, we only selected studies written in English or Spanish language, which could cause selection bias and impact in sports such as kabaddi, where non-English research may be more common). It is also important to acknowledge that the results of this overview are based on a limited number of sports, and that in many of them, there are no analyses separated into male and female athletes, despite the potential differences in the development of musculoskeletal injuries between the sexes. For some sports, exposure was reported in hours; in others, in athlete-exposures. While this is a limitation of the included reviews, it is also important readers are aware of this when interpreting and comparing the results across sports. Furthermore, in some sports, we found a high degree of overlap between the meta-analyses evaluated, so readers should be aware of this when interpreting the results of this overview.

## 5. Conclusions

Overall, ACL injuries may have a high incidence in American football, basketball, European football/soccer, and volleyball, among other sports. Sport-related ACL injuries show higher incidence rates in females than males across several sports. On the ither hand, prevalence was not meta-analyzed and emphasize that this remains a key gap in literature.

## Figures and Tables

**Figure 1 jfmk-10-00174-f001:**
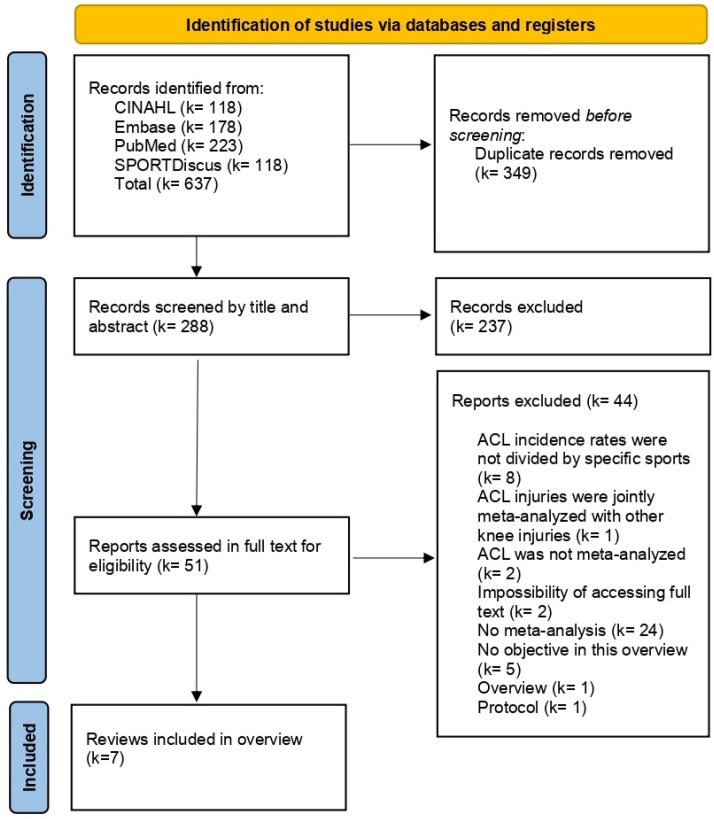
PRISMA 2020 flow diagram.

**Figure 2 jfmk-10-00174-f002:**
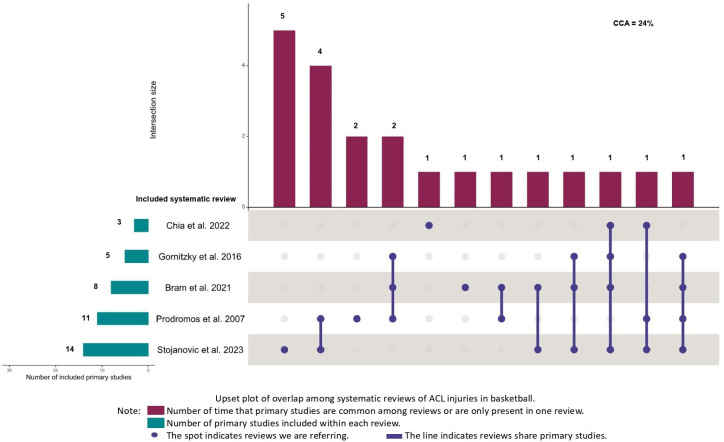
Upset plot of overlap among systematic reviews of ACL injuries in basketball [9,10,11,13,14].

**Figure 3 jfmk-10-00174-f003:**
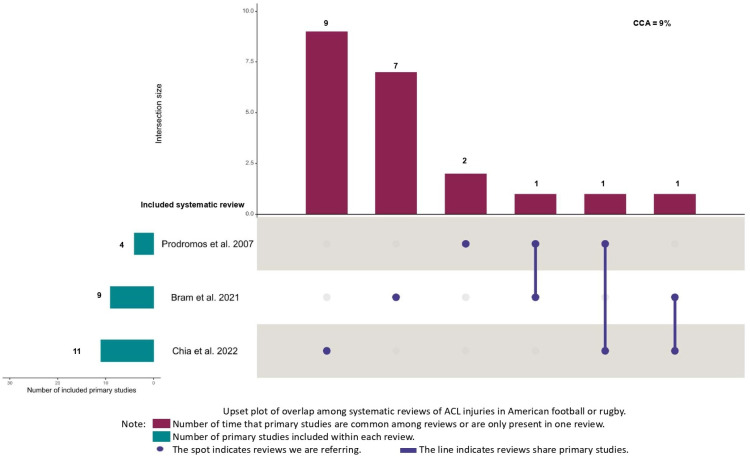
Upset plot of overlap among systematic reviews of ACL injuries in American football or rugby [9,10,13].

**Figure 4 jfmk-10-00174-f004:**
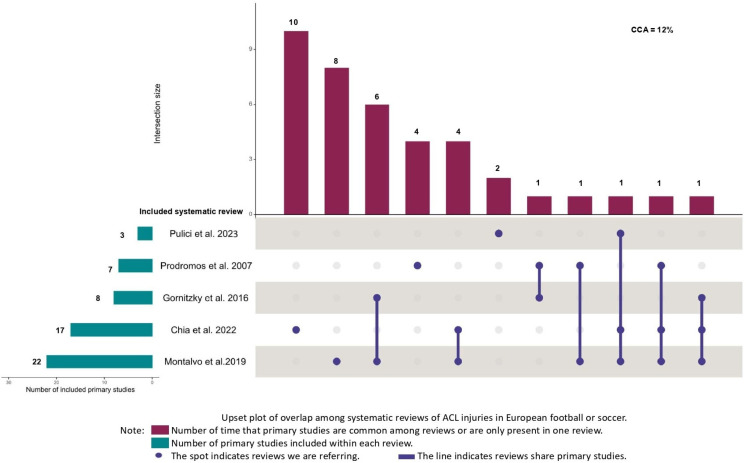
Upset plot of overlap among systematic reviews of ACL injuries in European football or soccer [1,10,11,12,13].

**Figure 5 jfmk-10-00174-f005:**
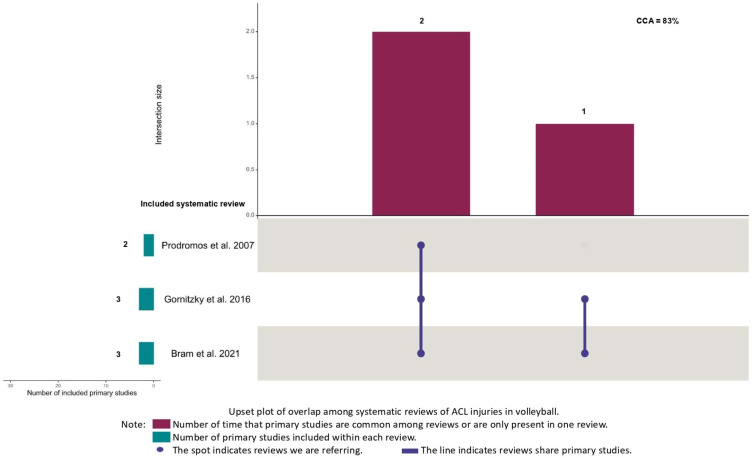
Upset plot of overlap among systematic reviews of ACL injuries in volleyball [9,11,13].

## Data Availability

The original contributions presented in this study are included in the article/Appendix A. Further inquiries can be directed to the corresponding author.

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
