# Peer review of "The Incidence of Sport-Related Anterior Cruciate Ligament Injuries: An Overview of Systematic Reviews Including 51 Meta-Analyses"

_jfmk, 2025, doi:10.3390/jfmk10020174_

Round 1
Reviewer 1 Report
Comments and Suggestions for Authors
The manuscript presents a well-organized and comprehensive overview of systematic reviews and meta-analyses evaluating the incidence rates of anterior cruciate ligament (ACL) injuries in sport-related contexts. The topic is highly relevant considering the clinical, economic, and social impact of ACL injuries across various sports disciplines. The authors appropriately employ rigorous methodological standards (PRIOR, PRISMA, AMSTAR 2) and prospectively register the review protocol, which strengthens the transparency and reliability of the work.
The manuscript is clear, methodologically sound, and clinically important, although some areas require minor improvement for optimization before publication.
The article fills an important gap by synthesizing a growing body of systematic reviews on ACL injury incidence. While the overview highlights incidence estimates well, the fact that no prevalence data was meta-analyzed warrants deeper reflection. It would strengthen the Discussion section to more explicitly propose strategies for addressing this gap in future research.
The study adheres to PRIOR and PRISMA guidelines, prospective protocol registration, and overlap analysis (CCA) using upset plots represent best practices, but the search was limited to studies published in English and Spanish. Although this is justified, a clearer acknowledgment of the potential bias introduced by language restrictions should be moved to the Limitations section rather than only briefly mentioned.
The matrix approach and CCA calculations are appropriate and visually well-represented. Some incidence ranges (e.g., for basketball) are quite broad, reflecting heterogeneity between included reviews. It would be valuable to briefly comment on possible sources of heterogeneity in the Results or Discussion sections.
The discussion appropriately contextualizes the findings with gender/sex disparities and geographic differences. Further critical interpretation is warranted concerning how different methodological qualities of included reviews (e.g., failure to use GRADE, lack of prospective protocols) might impact the overall confidence in conclusions. Currently, this point is touched on but not fully integrated into the broader implications.
The manuscript correctly points out the underuse of GRADE in included reviews. It would strengthen the manuscript to explicitly recommend the systematic application of GRADE in future epidemiological meta-analyses.
The writing is generally very good. The manuscript is readable and logical. However, some sentences in the Results section could be tightened to avoid redundancy ("Finally, one review combined female and male athletes..." and similar repetitive phrasings).
Figures are informative and appropriate. However, ensuring that figures are of high resolution and fully compliant with JFMK's style formatting (colors, labels, etc.) during the production phase is important.
Minor issues
Abstract: The Methods section redundantly mentions "Study design" and "Methods" — this could be streamlined into a single statement.
Introduction: Citations for statistics related to the economic burden should be updated in the final version if more recent data exist.
Methodological Deviations: Good that deviations from the protocol are transparently reported; however, a brief rationale for focusing only on athlete populations would improve justification.
Limitations Section: Should slightly expand on language restriction and heterogeneity acknowledgment.
Author Response
Dear Editor,
Thank you very much for allowing us to address all the important commentaries that the reviewers have provided. They have greatly improved the quality of this manuscript. See below our responses point-by-point. Thanks!
Reviewer 1
The manuscript presents a well-organized and comprehensive overview of systematic reviews and meta-analyses evaluating the incidence rates of anterior cruciate ligament (ACL) injuries in sport-related contexts. The topic is highly relevant considering the clinical, economic, and social impact of ACL injuries across various sports disciplines. The authors appropriately employ rigorous methodological standards (PRIOR, PRISMA, AMSTAR 2) and prospectively register the review protocol, which strengthens the transparency and reliability of the work. The manuscript is clear, methodologically sound, and clinically important, although some areas require minor improvement for optimization before publication. The article fills an important gap by synthesizing a growing body of systematic reviews on ACL injury incidence.
Response: Thank you very much for your positive feedback and your important suggestions that have improved our overview.
While the overview highlights incidence estimates well, the fact that no prevalence data was meta-analyzed warrants deeper reflection. It would strengthen the Discussion section to more explicitly propose strategies for addressing this gap in future research.
Response: We have developed now a specific subheading namely future research (discussion 4.1) where we have expanded this issue.
The study adheres to PRIOR and PRISMA guidelines, prospective protocol registration, and overlap analysis (CCA) using upset plots represent best practices, but the search was limited to studies published in English and Spanish. Although this is justified, a clearer acknowledgment of the potential bias introduced by language restrictions should be moved to the Limitations section rather than only briefly mentioned.
Response: This has been now included in the limitations section.
The matrix approach and CCA calculations are appropriate and visually well-represented. Some incidence ranges (e.g., for basketball) are quite broad, reflecting heterogeneity between included reviews. It would be valuable to briefly comment on possible sources of heterogeneity in the Results or Discussion sections.
Response: We have included now some information in the discussion section, before future research considering this suggestion.
The discussion appropriately contextualizes the findings with gender/sex disparities and geographic differences. Further critical interpretation is warranted concerning how different methodological qualities of included reviews (e.g., failure to use GRADE, lack of prospective protocols) might impact the overall confidence in conclusions. Currently, this point is touched on but not fully integrated into the broader implications.
Response: A further discussion about GRADE has been included now in the discussion.
The manuscript correctly points out the underuse of GRADE in included reviews. It would strengthen the manuscript to explicitly recommend the systematic application of GRADE in future epidemiological meta-analyses.
Response: Response: A further discussion about GRADE has been included now in the discussion.
The writing is generally very good. The manuscript is readable and logical. However, some sentences in the Results section could be tightened to avoid redundancy ("Finally, one review combined female and male athletes..." and similar repetitive phrasings).
Response: This has been corrected.
Figures are informative and appropriate. However, ensuring that figures are of high resolution and fully compliant with JFMK's style formatting (colors, labels, etc.) during the production phase is important.
Response: Thank you. We will send the editorial office full high-quality figures if they request us.
Minor issues Abstract: The Methods section redundantly mentions "Study design" and "Methods" — this could be streamlined into a single statement.
Response: Corrected.
Introduction: Citations for statistics related to the economic burden should be updated in the final version if more recent data exist.
Response: Thank you. We do not have found any new study considering this point.
Methodological Deviations: Good that deviations from the protocol are transparently reported; however, a brief rationale for focusing only on athlete populations would improve justification.
Response: This has been added.
Limitations Section: Should slightly expand on language restriction and heterogeneity acknowledgment.
Responses: This has been included now.
Reviewer 2 Report
Comments and Suggestions for Authors
You can find the comments atached.

Author Response
Reviewer 2
Thank you very much for your positive feedback and your suggestions that have greatly improved the quality of this overview.
- Abstract 1A. The abstract is informative and presents the objective, methods, and findings clearly. However, the aim could be phrased more precisely as: “to synthesize pooled incidence rates of sport-related ACL injuries based on published systematic reviews with meta-analyses.” Response: This has been changed according to your suggestion.
1B. Consider briefly including a sentence on how this overview can guide clinicians, researchers, or policymakers (e.g., gender-specific prevention or resource allocation).
Response: This has been included.
- Introduction 2A. The introduction provides a strong rationale for conducting this overview. The economic burden and clinical consequences of ACL injuries are well presented. The section is well supported by relevant references. However, although from lines 45–55, the text flows well but lacks citation support, particularly when stating the need for this overview and the absence of pooled prevalence data. Please include at least one reference supporting the current gap in prevalence-focused meta-analyses.
Response: The introduction has been revised.
2B. To support the applied and methodological importance of such evidence syntheses, consider citing: Schelling, X., Alonso-Perez-Chao, E., & Robertson, S. (2025). Implementation of a Decision Support System to Enhance Movement Proficiency Assessment in Sport. Journal of Functional Morphology and Kinesiology, 10(1), 86.
Response: This has been included.
